# Thermal Death Kinetics of Three Representative *Salmonella enterica* Strains in Toasted Oats Cereal

**DOI:** 10.3390/microorganisms10081570

**Published:** 2022-08-04

**Authors:** Matthew Chick, Antonio Lourenco, Alice Maserati, Ryan C. Fink, Francisco Diez-Gonzalez

**Affiliations:** 1Department of Food Science and Nutrition, University of Minnesota, Saint Paul, MN 55455, USA; 2Food Biosciences Department, Teagasc Food Research Centre, Moorepark, P61 C996 Fermoy, Ireland; 3Faculty of Computer Science, Dalhousie University, Halifax, NS B3H 4R2, Canada; 4Institute for Comparative Genomics, Dalhousie University, Halifax, NS B3H 4R2, Canada; 5Center for Food Safety, University of Georgia, 1100 Experiment St., Griffin, GA 30223, USA

**Keywords:** *Salmonella*, thermal resistance, inactivation, low-water activity, cereal

## Abstract

Several reports have indicated that the thermal tolerance of *Salmonella* at low-water activity increases significantly, but information on the impact of diverse food matrices is still scarce. The goal of this research was to determine the kinetic parameters (decimal reduction time, D; time required for the first decimal reduction, δ) of thermal resistance of *Salmonella* in a previously cooked low water activity food. Commercial toasted oats cereal (TOC) was used as the food model, with or without sucrose (25%) addition. TOC samples were inoculated with 10^8^ CFU/mL of a single strain of one of three *Salmonella* serovars (Agona, Tennessee, Typhimurium). TOC samples were ground and equilibrated to a_w_ values of 0.11, 0.33 and 0.53, respectively. Ground TOC was heated at temperatures between 65 °C and 105 °C and viable counts were determined over time (depending on the temperature for up to 6 h). Death kinetic parameters were determined using linear and Weibull regression models. More than 70% of Weibull’s adjusted regression coefficients ($R_{adj}^{2}$) and only 38% of the linear model’s $R_{adj}^{2}$ had values greater than 0.8. For all serovars, both D and δ values increased consistently at a 0.11 a_w_ compared to 0.33 and 0.53. At 0.33 a_w_, the δ values for Typhimurium, Tennessee and Agona were 0.55, 1.01 and 2.87, respectively, at 85 °C, but these values increased to 65, 105 and 64 min, respectively, at 0.11 a_w_. At 100 °C, δ values were 0.9, 5.5 and 2.3 min, respectively, at 0.11 a_w_. The addition of sucrose resulted in a consistent reduction of eight out of nine δ values determined at 0.11 a_w_ at 85, 95 and 100 °C, but this trend was not consistent at 0.33 and 0.53 a_w_. The Z values (increase of temperature required to decrease δ-value one log) were determined with modified δ values for a fixed β (a fitting parameter that describes the shape of the curve), and ranged between 8.9 °C and 13.4 °C; they were not influenced by a_w_, strain or sugar content. These findings indicated that in TOC, high thermal tolerance was consistent among serovars and thermal tolerance was inversely dependent on a_w_.

## 1. Introduction

Due to its pervasive presence and its tolerance to different stresses, *Salmonella* can easily contaminate food products and processing plants. *Salmonella* is a frequent contaminant of animal origin foods, such as beef, poultry, pork and eggs [1,2,3,4]; it also contaminates fish, shrimp and dairy products [5]. Most of these foods are high-water activity (a_w_) products. In recent years, however, foodborne disease caused by *Salmonella* has also been linked to low a_w_ foods such as grain flours, raw and processed nuts, dried milk, black and red pepper, peanut butter, rice and animal feeds [6,7,8,9,10,11,12,13,14].

Epidemiological reports confirmed that in 2015, *Salmonella* was the leading bacterial cause of foodborne disease in the United States; it was responsible for 7728 cases, 2074 hospitalizations and 32 deaths [15]. In Europe, the latest EFSA report points out that there were 52,702 confirmed cases of human salmonellosis for the year 2020 [16]. *Salmonella* causes more than 100 outbreaks of foodborne disease in the US annually, and most of these are caused by poultry and eggs [17]. The increasing number of salmonellosis outbreaks associated with multiple low a_w_ foods has prompted the industry to adopt proactive measures and good practices, including enhanced surveillance and food testing [18].

The reduction of available water, especially though drying, has been a well-established strategy to control bacteria [19], but the increased incidence of low-moisture food-related outbreaks suggests that *Salmonella* is capable of surviving for long periods of time in dry foods and in low-water activity matrices [20,21]. Many of the foods associated with *Salmonella* outbreaks are often subjected to heat treatments, such as baking or roasting. This is the case, for example, for puffed cereals, peanut butter or chocolate; however, these foods have been associated with outbreaks [9,10,22], suggesting that thermal treatment of these food products may not be sufficient to kill *Salmonella* [13,23,24,25]. Different factors that influence *Salmonella* heat tolerance in low water activity foods might include the intrinsic structural property of the food matrix, the presence of different microenvironments [26], and the specific composition of the food, such as its salt, sugar and fat content [27,28,29,30]. In the case of toasted oats cereals, manufacturers often include commercial variants that contain sugar.

Some of the studies that determined kinetic parameters of *Salmonella* inactivation in low a_w_ foods included those concerning dry corn flour, almonds, peanut butter, flour, cocoa, hazelnut and whey protein powder shells [23,24,25,29,31,32]. On dry corn flour, decimal reduction times (D-value) at 49 °C of eight different serovars of *Salmonella* varied from 0.3 to 9.9 h [24], while D-values at 100 °C for *S.* Oranienburg and Enteritidis were around 2.5 min in cocoa and between 7 and 11 min in hazelnuts (both at 4% moisture). An increase in moisture (up to 7% moisture) markedly affected their thermal tolerance to D-values at 80 °C for between 5.4 and 7.7 in cocoa and between 2.5 and 4.5 min in hazelnuts. These findings clearly indicated that this pathogen is especially difficult to kill in low a_w_ foods.

In order to study *Salmonella*’s thermal response under these conditions, the traditional linear inactivation model [33] as well as non-linear models such as the Weibull model [34] are often applied. The development of predictive models often requires a comparison of the statistical parameters associated with these models [35]. For low-water-activity foods, recent studies have indicated that a single predictive model does not always fit every condition tested [36]. These studies reported that the Weibull distribution generally provides a better fit than the linear model, but its correct use depends on the experimental design [37].

Despite the occurrence of two salmonellosis outbreaks in toasted oats cereal products in 1998 and 2008 [6,9], this food matrix has never been investigated as a food matrix to determine inactivation kinetics of *Salmonella*. This study was undertaken to determine the effect of low-water activity on a toasted oats cereal (TOC) matrix, and to evaluate the impact of sugar addition on the thermal inactivation kinetics of three *Salmonella enterica* subsp. *enterica* strains of different serovars.

## 2. Materials and Methods

### 2.1. Strains and Culture Preparation

The strains of *Salmonella enterica* subsp. *enterica* serovars used in this research were *S.* Typhimurium E2009005811, *S.* Tennessee E200700502 and *S.* Agona. The first two strains were provided by the Minnesota Department of Health; they were isolated from patients linked to the *S.* Typhimurium E2009005811 and *S.* Tennessee E200700502 peanut butter outbreaks of 2009 and 2007, respectively [8,10]. The *S.* Agona strain was originally isolated from an outbreak related to toasted oats cereal from 1998 [6]. The stock cultures of the three serovars were stored in a 1:1 ratio of glycerol and tryptic soy broth (TSB; Neogen, Inc., East Lansing, MI, USA) at −55 °C. The working cultures of each serovar were prepared from frozen stocks and inoculated into TSB, grown overnight at 37 °C and then stored at 4 °C. In order to test the working stocks, they were re-transferred once a week and streaked onto tryptic soy agar (TSA; Neogen, Inc.) containing 0.8 g/L ferric ammonium citrate (Sigma-Aldrich™, St. Louis, MO, USA) and 6.8 g/L sodium thiosulfate. (Acros Organics, Morris Plains, NJ, USA). This medium was formulated to provide a differential but non-selective agar. Periodically, the serovars were also streaked onto bismuth sulfate agar (Neogen, Inc.) and xylose lysine deoxycholate agar (Neogen, Inc.).

### 2.2. Inoculation and Drying Procedure

From working stocks, inoculation cultures were incubated overnight in 40 mL of TSB at 37 °C, and added to bottles containing 360 mL of sterile water; they were gently shaken until a final count of approximately 10^8^ CFU/mL was obtained. Twenty grams of a commercially available TOC brand were added to the bottles and mixed by repeated inversion for one minute. TOC samples had an initial (immediately after opening the package) a_w_ value that ranged from 0.18 to 0.30. The inoculated cereal samples were separated with sterile kitchen strainers and spread out on sterile perforated kitchen baking sheets. The baking sheets with cereal were placed in an incubator at 40 °C for 12 to 18 h, in order to facilitate drying. The final weights of TOC samples were verified to be 20.0 ± 0.5 g. The cereal samples were then ground using a sterilized mortar and pestle in a biosafety cabinet, and then placed on aluminum foil trays before water activity equilibration. The particle size of ground TOC was less than 1 mm.

For the treatments involving added sucrose, samples of 7.5 g of commercial food grade powdered sucrose (Domino, Yonkers, NY, USA) were added to cereals that were spread onto baking sheets. Sucrose was mixed into the cereal by folding in small increments, and then mixing the sucrose into the wet cereal with sterile spoons. Once trays were removed from the incubator at the end of the drying period, they were weighed to ensure that TOC samples had a total weight of 26.5 ± 0.5 g, or approximately 25% by weight of sucrose. Sucrose-containing cereal was also ground using the same procedure as described above.

### 2.3. Preparation of Samples for Thermal Inactivation

A total of 26.5 g of TOC or 20 g TOC without sucrose was separated and placed onto separate foil trays. TOC samples were equilibrated to specific water activities by storage for periods between 7 to 12 d in desiccators that contained saturated solutions of lithium chloride (Acros Organics), magnesium chloride (Sigma-Aldrich™) and magnesium nitrate (Acros Organics), in order to attain equilibrium at 0.11, 0.33 and 0.53 a_w_, respectively. After the storage period, trays containing the cereal were removed and the water activity was measured in order to ensure that the samples were within 0.02 a_w_ of the target value of the specific desiccator that the samples were put into. Water activity of the TOC samples was measured using a water activity meter (Pawkit Model, Decagon Devices, Inc., Pullman, WA, USA) that was calibrated every other day according to the manufacturer’s procedure. If the cereal was not within 0.02 a_w_ of the target water activity within 7 to 12 days, the sample was not used.

Sterile 12-cubic-centimeter syringes were used to fill capillary tubes (1.5–1.8 × 90 mm borosilicate glass) with ground TOC by inserting the tubes through the Luer-lock tip of the syringes. Sterile ram rods (118 mm × 1 mm stainless steel) were used to completely fill the capillary tubes when necessary. The TOC-filled tubes were heat sealed using a propane hand torch and placed in a solution of 10% commercial chlorine bleach (5.25% sodium hypochlorite concentration) for at least one minute, in order to sterilize the exterior of the tubes. The average weight of TOC in the capillary tubes was 0.05 g. In order to assure that the water activity of the cereals had not changed, after all capillary tubes had been sterilized, a_w_ was measured using the remaining cereal in the syringe. If the remaining cereal’s water activity varied by more than 0.02 a_w_, none of the capillary tubes were used. For all tested conditions, three strains, three temperatures, three water activities, with and without sucrose were used, and at least two independent experiments were performed.

### 2.4. Thermal Inactivation

All sealed capillary tubes were placed into either an oil bath (High Temp Bath 160 A, Fisher Scientific, Inc., Waltham, MA, USA) or water bath (Isotemp 205, Fisher Scientific, Inc.) depending on the temperature tested, and were calibrated once a month. The water baths were set to the testing temperatures of between 60 to 95 °C, while the oil bath temperatures were set to between 85 and 105 °C. Two capillary tubes were removed at predetermined time intervals and immediately placed in an ice bath for one minute. From the ice bath, the tubes were placed in a solution of 10% bleach and rinsed with sterile water. Individual capillary tubes were then placed in separate, sterilized 24 × 150 mm screw cap test tubes, each containing a magnetic stir bar (25 × 5 mm); they were vortexed until the capillary tubes were ground to release their contents. Ten milliliters of phosphate buffer (PB) were added to the test tubes and mixed for 10 s. These buffer suspensions were serially diluted by transferring 1 mL serially into 9-milliliter PB tubes. Volumes of 0.1 mL from each dilution were spread plated in duplicate, on TSA containing 0.8 g/L ferric ammonium citrate and 6.8 g/L sodium thiosulfate. This growth medium was intended to address the possibility of cell injury caused by heat, instead of using a standard selective *Salmonella* medium. The plates were incubated for 24 h at 37 °C before counting colonies was performed. Non-inoculated TOC samples were routinely tested to determine the count of naturally present organisms capable of producing black precipitate on modified TSA. None of those control samples was positive for hydrogen sulfide-producing microorganisms (detection limit of 100 CFU/g).

The counts of surviving cells were calculated using the aerobic plate count formula from the Food and Drug Administration’s Bacteriological Analytical Manual, and adjusted for a 0.1-milliliter plating volume (Maturin and Peeler, 1998). Each capillary tube sample testing was calculated individually and averaged with its replicates.

After inactivation, the following two inactivation models were fitted using Microsoft Excel 2016 ad-In GInaFiT Version 1.6 (Geeraerd, Valdramidis, and Van Impe, 2005).

The log-linear model [33] used is shown below:(1)$$N_{t}=N_{0}.e^{\left(-k_{max}.t\right)}$$
where $N_{t}$ is the population at time *t* (CFU/g), $N_{0}$ is the population at time 0 (CFU/g),$\;k_{max}$ is the maximum specific inactivation rate (min^−1^) and the $D_{value}=\frac{ln10}{k_{max}}$.

The Weibull model [34] used is shown below:(2)$$log\left(N_{t}\right)=log\left(N_{0}\right)-{\left(\frac{t}{\delta }\right)}^{\beta }$$
where $N_{t}$ and, $N_{0}$ are as previously described, δ is the time required for the first decimal reduction (min) and β is a fitting parameter that describes the shape of the curve (β>1 convex, β<1 concave).

In order to evaluate the goodness-of-fit of the two models to the inactivation data, the adjusted coefficient of determination ($R_{adj}^{2})$ Equation (3), the $f_{value}$ $\left(f_{test}\right)$ (Equations (6) and (7)), the root mean square error (RMSE) (Equation (13)) and the corrected Akaike information criterion (AIC_c_) (Equation (15)) were calculated according to the formulas shown below, for which n is the total number of observations at all time points, m is the number of time points, p is the number of parameters in the model and k=p+1.
(3)$$R_{adj}^{2}=1-\frac{\left(n-1\right)\left(1-R^{2}\right)}{df_{model}}$$
where
(4)$$R^{2}=\frac{\sum {\left(logN_{model}-log{\bar{N}}_{data}\right)}^{2}}{\sum {\left(logN_{model}-log{\bar{N}}_{data}\right)}^{2}+\sum {\left(logN_{model}-logN_{data}\right)}^{2}}$$
(5)$$df_{model}=n-p$$
(6)$$f_{test}=\frac{MSE_{model}}{MSE_{data}}$$
and
(7)$$F_{table}=\frac{df_{model}}{df_{data}}$$
where
(8)$$MSE_{model}=\frac{RSS_{model}}{df_{model}}$$
(9)$$RSS_{model}=\sum {\left(logN_{model}-logN_{data}\right)}^{2}$$
(10)$$MSE_{data}=\frac{RSS_{data}}{df_{data}}$$
(11)$$RSS_{data}=\sum {\left(log\bar{N}-logN_{data}\right)}^{2}$$
(12)$$df_{data}=n-m$$
(13)$$RMSE=\sqrt{\frac{RSS_{model}}{df_{model}}}$$
where
(14)$$RSS_{model}=\sum {\left(logN_{model}-logN_{data}\right)}^{2}$$
(15)$$AIC_{c}=n\;ln\left(\frac{RSS_{model}}{n}\right)+2k+\frac{2k\left(K+1\right)}{n-k-1}$$

The Pearson correlation coefficient was calculated for each strain using Microsoft Excel 2016, in order to evaluate the correlation between temperatures, a_w_ and sugar on the inactivation parameters δ and β. As a result of the strong correlation between the δ and β values (Couvert, O., Gaillard, S., Savy, N., Mafart, P. and Leguérinel, 2005) the β value was fixed, which allowed us to compare the δ-values and further calculate Z-values for the first decimal reduction (the increase in temperature required to decrease δ-value by one log). For a given strain at each a_w_, the mean of the β values that was obtained at each temperature for which the data passed the F test (95% confidence interval) was used to obtain a value for the fixed β. The δ-value was then re-estimated using this value.

The *Z* value was then calculated according to Equation (16) shown below, where $\delta^{*}$ is the first decimal reduction for temperature $T^{*}$:(16)$$log\delta =log\delta^{*}-\left(\frac{T-T^{*}}{Z}\right)$$

Differences between means of δ-values were determined using Student’s *t*-test with a *p* < 0.05 significance.

## 3. Results

When inoculated samples of TOC were thermally treated, the D- and δ-values obtained for each of the three strains decreased as the temperature increased; however, the extent of the decline was affected by water activity (Figure 1). At an a_w_ of 0.11 and 85 °C, the D- and δ-values for the strains ranged from 148 to 201 min and from 64 to 105 min, respectively. The same thermal treatment using the samples that were pre-incubated at a_w_ 0.33 resulted in a significantly lower range of D- and δ-values of 6–22 min and 0.5–2.9 min, respectively. The effect of low-water activity increasing thermal inactivation rates was not consistently observed at 0.33 a_w_ as compared to 0.53. At 75 °C, δ-values ranged from 16 to 29 min at a_w_ 0.33, and from 5 to 18 min at a_w_ 0.53. The extent of this overlap was greater at 80 °C, with calculated δ-values of 2, 2.3 and 9 min at a_w_ 0.33, and 0.2, 3.2 and 3.6 min at a_w_ 0.53. These values were comparable to δ-values obtained at 100 °C of 0.9, 5.4 and 5.5 min, but at an a_w_ of 0.11 (Figure 1, Appendix A).

In plain TOC, out of 36 individual treatments (3 strains × 3 a_w_ × 4 temperatures), 33 treatments had $R_{adj}^{2}$ values that were greater for the Weibull model than for the linear model (Figure 1, Appendix A). When the linear and Weibull models were fitted to the thermal inactivation data, it was possible to observe that the Weibull model also fit the data better, with a smaller root mean square error (RMSE) and smaller corrected Akaike information criteria (AICc) values for all three strains (Table 1). However, for some conditions, the data did not pass the F test (95% confidence interval) as shown in Appendix A, where the parameters used to generate Figure 1 are shown; those conditions are marked with an asterisk. Overall, the δ-values of the three strains obtained by fitting the Weibull model increased substantially as the water activity declined. This effect is, however, much more evident from a_w_ 0.33 to a_w_ 0.11 than it is from a_w_ 0.53 to a_w_ 0.33.

When inoculated samples of TOC were supplemented with 25% sucrose, the D- and δ-values obtained for each of the three strains decreased as the temperature increased (Figure 2). At an a_w_ of 0.11 and 85 °C, the D- and δ-values for the strains ranged from 45 to 65 min and from 12 to 22 min, respectively. These values were significantly lower (*p* < 0.05) than the equivalent values obtained without sucrose at the same temperature–a_w_ combination. The same trend was observed at 95 °C and 100 °C at 0.11 a_w_. At higher water activity levels, the addition of sucrose did not consistently affect the D- and δ-values for the three strains.

In the samples that contained sucrose and were incubated at an a_w_ value of 0.33 and heated at 85 °C, the D- and δ-values were no more than 10% and 5%, respectively, of the same values determined at 0.11 a_w_ (Figure 2, Appendix A). In contrast to plain TOC samples, however, for all three strains the δ-values were consistently greater at 0.33 a_w_ in comparison to 0.53 a_w_ (*p* < 0.05). At 0.33 a_w_, the average δ-values were 32.3, 11.9 and 3.6 min at 75, 80 and 85 °C, respectively, while at 0.53 a_w_, corresponding δ-values were 8.3, 2.9 and 0.39 min.

The Weibull fitting for all TOC treatment samples with 25% sucrose resulted in $R_{adj}^{2}$ values greater than 0.84 (Appendix A). For all treatments, $R_{adj}^{2}$ values for the Weibull model were consistently higher than for the linear model. The better fit of the Weibull model was also corroborated with smaller RMSE and smaller AICc values for all three strains (Table 2).

Table 3 shows the Pearson correlation coefficients for the relationship of a_w_ and sugar, and their effect on the inactivation parameters δ and β.

When the change in temperature required to reduce the δ-value by one log (Z value) for the first log reduction was calculated using a log-linear model for most of the strains in all conditions, the adjusted $R^{2}$ values were high (greater than 0.96) for 16 out of 18 conditions. However, the obtained Z values did not change much either with the water activity, the strain or with the addition of sucrose to the TOC. The values ranged from 8.86 °C to 12.04 °C in the absence of sucrose, and from 9.47 °C to 13.45 °C in the presence of sucrose (Table 4).

## 4. Discussion

Although *Salmonella enterica* is one of the top causative agents of foodborne diseases, the mechanisms by which this pathogen enters and survives through the food production chain and survives food processing are still unclear, especially for low-moisture foods. Under low-water activity conditions, the food matrix, storage conditions as well as duration of storage have been reported to influence *Salmonella*’s ability to survive [38]. Several reports have shown that the low a_w_ typical of dry foods such as peanut butter, flour, cocoa, almonds, hazelnut and spices can also enhance the thermal tolerance of *Salmonella* cells [23,24,25,31,32,39]. The current study assessed the heat tolerance of three different *Salmonella* serovars that were previously isolated from outbreaks linked to low water activity foods, at three different low-water activities, with and without the presence of sucrose, for four temperature treatments, using a commercially available toasted oats breakfast cereal matrix which is similar to one of the products associated with the outbreaks.

Our study identified an inverse relationship between heat resistance of *Salmonella* and low-water activity in a food matrix. The inverse correlation between heat resistance and water activity appears to be greater in *Salmonella* than in other organisms reported in the literature [24,40,41]. We observed very similar thermal inactivation kinetic parameters for three *Salmonella* strains that were isolated from different foods. Although not significantly different, *S.* Agona seemed to be more heat tolerant among the strains analyzed, since it had the highest δ-values in at least 10 out of 24 temperature–a_w_ combinations compared to the other serovars. This observation is in agreement with a previous study performed by Santillana Farakos et al. [29]. That study also reported a greater tolerance of *S.* Agona along with *S.* Tennessee during a two-day storage challenge at 70 °C using a cocktail of serovars as the inoculum. Even in this case, there was no significant difference found between the two strains. However, VanCauwenberge et al. [24] observed the largest D-value for *S.* Tennessee in 15% moisture flour at 49 °C when it was compared to nine other *Salmonella* serovars.

As indicated above, several researchers have reported that survival rates increase as a_w_ values decrease, but the mechanisms involved in this response have yet to be fully elucidated. It can be hypothesized that possible mechanisms such as the increased stabilization of ribosomes, the influence of small amounts of osmoprotectants, the induction of viable but nonculturable states of bacterial cells, and the coagulation or oxidation of proteins in microbial cells may be determinants of survival [42]. Specifically, a global gene regulator such as RpoS has also been linked to increased survival rates at lower a_w_ [43]. The role of noncoding DNA and RNA has also been proposed as a protective mechanism by a few authors [44,45].

Thermal inactivation kinetics of *Salmonella* in high-moisture foods have been extensively studied, but research intended to determine inactivation kinetics in extruded cereal foods is relatively limited. Those reports of *Salmonella* inactivation in low a_w_ foods have investigated different matrices that included peanut butter, flour, pet food, cocoa, almonds and hazelnut shells. Our study is unique for using TOC because a commercial TOC product was previously involved in an outbreak. This study is also one of the few studies that have investigated thermal tolerance at one of the lowest a_w_ values. Other studies have measured the thermal resistance of *Salmonella* serovars below 0.5 a_w_, but on different matrices such as peanut butter (approx. 0.45 a_w_) [23,46,47]. The heat resistance of *Salmonella* has also been determined in dry carbohydrate-based foods in two separate studies that used corn or wheat flour [24,31].

Some reports have investigated the thermal inactivation of *Salmonella* at temperatures of 90 °C or higher. D-values at 90 °C from 9 to 13 min at 0.45 a_w_ [23] and from 4 to 7 min at 0.2 a_w_ were determined in peanut butter [46]. In TOC with sucrose, the D-values ranged from 15 to 24 min at 0.11 a_w_, and from 3.6 to 4.4 min at 0.33 a_w_ at the same temperature. At 100 °C, D-values of 2.5 and 7 to 11 min were determined in cocoa bean shells and hazelnut shells, respectively, using *S.* Enteritidis, *S.* Montevideo, *S.* Napoli, *S.* Oranienburg, *S.* Poona, *S.* Senftenberg and *S.* Typhimurium [32]. At the same temperature, we measured D-values in plain TOC that ranged from 13 to 22 min at 0.11 a_w_.

Two studies that evaluated heat tolerance in flours utilized methods that were different than the approach used in this investigation. Both studies used dry air as a heat source while the inoculated flour was spread into thin layers on foil trays. Only one of those studies measured the correlation between *Salmonella* thermal tolerance and changes in water activity; the authors of that study could not find a clear trend [31]. In contrast, we found that δ-values increased as the water activity was lowered to 0.11 a_w_.

In our study, the sensitivity of *Salmonella* to temperature changes, defined by the Z-value for the first log reduction, did not change under the conditions tested. In high-moisture foods, such as ground beef and chicken, several studies have reported Z-values of 10 ± 4 °C [48,49]. Studies that are focused on estimating Z-values for low-water activity food matrices are relatively rare. However, two such studies reported large Z-values of 39 to 56 °C in peanut butter, and of 15 to 54 °C in corn flour [23,24]. In our study, the average Z-values for the three serovars was consistently between 8.9 and 13.4 °C for TOC with and without sucrose. These values agree with previously published research that reported a Z-value of 8.3 °C in almonds [25].

It has been hypothesized that the food matrix composition can affect the thermal resistance of *Salmonella*, and some studies have investigated the impact of salt on thermal tolerance [29]. However, studies on the impact of sugar are more limited, and are generally focused on the long-term survival of *Salmonella* during storage rather than on its thermal resistance [50,51]. The observation that sucrose enhanced heat resistance of the bacterial proteins in different solutions has led to the hypothesis that this carbohydrate could also stabilize proteins in the cell, especially as water activity is lowered [52,53,54]. In hyperosmotic environments, microorganisms can produce or uptake certain molecules, referred to as ‘osmolytes,’ that counteract the effects of high osmotic pressure [55,56,57]. While glycerol and other polyols have been shown to increase heat resistance, sucrose has been reported to be the most effective [53].

The metabolic and genetic components of *Salmonella*’s ability to survive desiccation, in addition to its thermal tolerance abilities, have yet to be fully elucidated, but recent publications have identified that global regulators and specific components are involved. A study in chicken litter reported significant up-regulation of *rpoS* in *Salmonella* cells adapted to desiccation upon 3, 12, and 24 h [58]. All rpoS mutants exhibited a decreased tolerance to heat, compared to desiccated and non-desiccated wild types, suggesting that mutations in rpoS could lead to the loss of thermal tolerance in *Salmonella*. Maserati et al. showed that two genes involved in type III secretion systems and previously identified as virulence factors, sopD and sseD, were overexpressed in cells that were subjected to very low water activity [59]. Those genes were necessary for survival during desiccation, as the viability in low a_w_ of knockout mutants was markedly reduced compared to wild-type strains. These recent advances suggest that the response of *Salmonella* to such adverse conditions may be quite complex.

This study measured the effects that sucrose has on the heat resistance of *Salmonella* in low-water activity environments. Despite the lack of a clear trend on the δ-value observed for the overall water activities with or without sucrose, at a water activity of 0.11 we measured consistently shorter inactivation times than what we observed in the controls. This observation suggests that the protective effect of sucrose might only be relevant at higher water activities. In fact, our results corroborate the trends observed in previous reports that observed that sucrose could have a protective effect on cells’ viability for high-water activity. In TSB media with 35% (*w*/*w*) sucrose (a_w_ = 0.95), Peña-Meléndez et al. [60] observed a protective effect on three *Salmonella* serovars under both adaptation and osmotic shock, rendering the highest D_55°C_-values when compared to the other humectants tested (glycerol and NaCl).

The findings of the present study support the validation and the development of more effective processing of dry foods, in order to improve commercial high-temperature processes and ultimately assure the food safety of low-moisture foods.

## Figures and Tables

**Figure 1 microorganisms-10-01570-f001:**
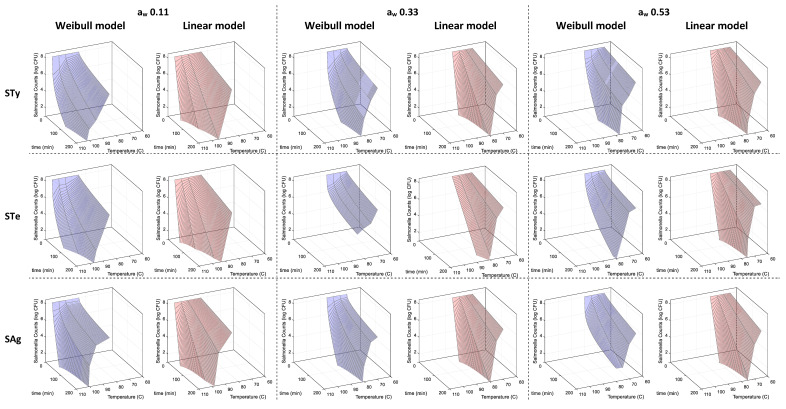
Thermal inactivation rates of *Salmonella enterica* serovars Typhimurium, Tennessee and Agona in toasted oat cereal (TOC affected by water activity (a_w_) and temperature). The Weibull model fitting is represented in blue and the linear model fitting is represented in red. STy, STe and SAg are abbreviations for serovars Typhimurium, Tennessee and Agona, respectively.

**Figure 2 microorganisms-10-01570-f002:**
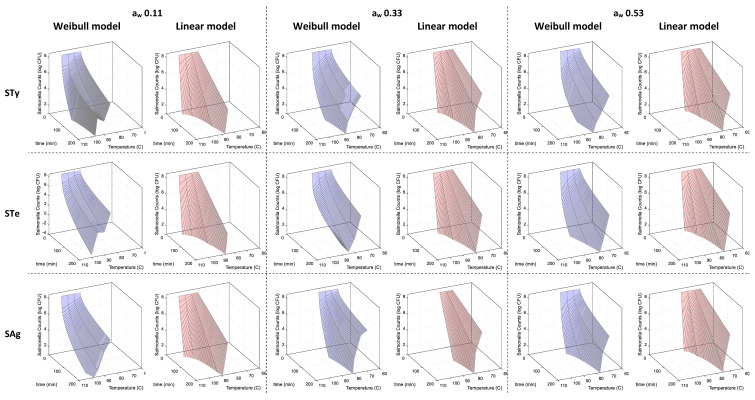
Thermal inactivation rates of *Salmonella enterica* serovars Typhimurium, Tennessee and Agona in toasted oats cereal (TOC) affected by water activity (a_w_) and temperature in the presence of 25% sucrose. Weibull model fitting is represented in blue and linear model fitting is represented in red. STy, STe and SAg are abbreviations for serovars Typhimurium, Tennessee and Agona, respectively.

**Table 1 microorganisms-10-01570-t001:** Evaluation of the goodness-of-fit of the linear and Weibull models used to describe the inactivation of three *Salmonella enterica* serovars by means of the root mean square error (RMSE) and the corrected Akaike information criterion (AIC_c_) calculated for each water activity and temperature in toasted oats cereal (TOC).

a_w_	Temp (°C)	Serovar Typhimurium				Serovar Tennessee				Serovar Agona			
		Linear Model		Weibull Model		Linear Model		Weibull Model		Linear Model		Weibull Model	
		RMSE	AIC_c_	RMSE	AIC_c_	RMSE	AIC_c_	RMSE	AIC_c_	RMSE	AIC_c_	RMSE	AIC_c_
**0.11**	**85**	0.44	−95.7	0.39	−110.1	0.39	−82.8	0.38	−83.9	0.58	−43.3	0.54	−48.6
	**95**	0.59	−54.4	0.50	−69.8	0.66	−37.0	0.53	−56.2	0.58	−42.9	0.57	−43.4
	**100**	0.70	−16.3	0.42	−44.3	0.64	−46.7	0.60	−51.9	0.48	−62.0	0.37	−81.8
	**105**	0.79	−9.5	0.46	−37.1	0.81	−17.8	0.65	−37.9	0.67	−29.8	0.67	−28.9
**0.33**	**70**	0.52	−70.1	0.51	−71.7	0.35	−46.9	0.29	−54.6	0.60	−44.7	0.52	−57.3
	**75**	0.54	−69.5	0.42	−97.2	0.80	−10.2	0.71	−16.1	0.57	−54.5	0.46	−75.7
	**80**	0.80	−26.1	0.62	−58.9	0.96	0.1	0.75	−19.0	0.68	−25.6	0.58	−37.2
	**85**	0.57	−33.7	0.24	−90.5	1.19	9.6	0.79	−1.5	0.54	−40.2	0.44	−52.0
**0.53**	**65**	0.16	−98.1	0.17	−95.7	0.81	−5.0	0.81	−3.3	0.64	−47.7	0.63	−47.5
	**70**	0.41	−63.6	0.37	−70.0	0.90	−2.5	0.89	−1.7	0.37	−50.8	0.35	−51.1
	**75**	0.61	−36.6	0.45	−59.0	0.65	−54.3	0.51	−86.3	0.27	−102.8	0.22	−120.1
	**80**	0.67	−28.3	0.64	−30.6	1.02	4.5	0.99	4.4	1.12	10.9	0.93	−0.1

**Table 2 microorganisms-10-01570-t002:** Evaluation of the goodness-of-fit of the linear and Weibull models used to describe the inactivation of three *Salmonella enterica* serovars by means of the root mean square error (RMSE) and the corrected Akaike information criterion (AIC_c_) calculated for each water activity and temperature in toasted oats cereal (TOC) containing 25% sucrose.

a_w_	Temp (°C)	Serovar Typhimurium				Serovar Tennessee				Serovar Agona			
		Linear Model		Weibull Model		Linear Model		Weibull Model		Linear Model		Weibull Model	
		RMSE	AIC_c_	RMSE	AIC_c_	RMSE	AIC_c_	RMSE	AIC_c_	RMSE	AIC_c_	RMSE	AIC_c_
**0.11**	**85**	0.32	−82.31	0.25	−100.86	0.39	−39.25	0.25	−57.87	0.30	−91.98	0.17	−138.29
	**90**	0.35	−72.63	0.23	−100.32	0.42	−58.72	0.35	−70.67	0.36	−77.42	0.17	−132.02
	**95**	0.47	−53.59	0.25	−99.56	0.44	−42.37	0.29	−63.98	0.32	−91.47	0.18	−139.94
	**100**	0.34	−72.42	0.19	−110.09	0.42	−41.93	0.29	−59.88	0.36	−80.06	0.23	−115.03
**0.33**	**75**	0.38	−73.39	0.24	−109.65	0.36	−76.28	0.29	−90.89	0.29	−97.08	0.17	−135.70
	**80**	0.47	−52.16	0.31	−81.03	0.39	−64.60	0.32	−77.80	0.41	−69.18	0.27	−102.86
	**85**	0.44	−65.40	0.25	−112.97	0.52	−46.29	0.40	−64.06	0.36	−81.47	0.26	−108.48
	**90**	0.50	−53.01	0.24	−113.14	0.47	−78.79	0.30	−127.15	0.34	−87.33	0.16	−148.92
**0.53**	**70**	0.39	−49.57	0.31	−60.46	0.36	−80.58	0.30	−93.22	0.45	−58.76	0.38	−70.88
	**75**	0.54	−64.66	0.31	−123.68	0.42	−64.66	0.28	−94.47	0.42	−45.41	0.15	−101.31
	**80**	0.61	−36.6	0.45	−59.0	0.44	−51.40	0.29	−77.54	0.55	−30.26	0.44	−40.94
	**85**	0.67	−28.3	0.64	−30.6	0.48	−55.46	0.23	−114.29	0.55	−30.42	0.43	−41.72

**Table 3 microorganisms-10-01570-t003:** Pearson (**ρ**) coefficients of the correlation between temperatures, a_w_ and sucrose content, with the inactivation parameters δ and β of toasted oats cereal (TOC) with and without sucrose.

TOC		*S.* Typhimurium			*S.* Tennessee			*S.* Agona		
	a_w_	ρ	Lower 95% CI	Upper 95% CI	ρ	Lower 95% CI	Upper 95% CI	ρ	Lower 95% CI	Upper 95% CI
Control	0.11	0.667	−0.819	0.992	0.825	−0.657	0.996	−0.546	−0.988	0.873
	0.33	0.984	0.431	1.000	0.963	0.022	0.999	−0.955	−0.999	0.080
	0.53	0.900	−0.451	0.998	0.441	−0.903	0.985	0.684	−0.809	0.993
25% Sucrose	0.11	0.691	−0.804	0.993	0.354	−0.920	0.9812	0.853	−0.600	0.997
	0.33	0.809	−0.683	0.996	0.634	−0.837	0.9911	0.168	−0.946	0.972
	0.53	0.866	−0.560	0.997	0.784	−0.718	0.9952	0.411	−0.909	0.984

**Table 4 microorganisms-10-01570-t004:** Change in temperature needed to obtain a 90% reduction in δ-values (Z-value, °C) for the first log reduction of *Salmonella* in toasted oats cereal (TOC), as affected by water activity (a_w_) and sucrose addition.

a_w_	Toasted Oats Cereal						Toasted Oats Cereal with 25% Sucrose					
	STy		STe		SAg		STy		STe		SAg	
	Z	$R_{adj}^{2}$	Z	$R_{adj}^{2}$	Z	$R_{adj}^{2}$	Z	$R_{adj}^{2}$	Z	$R_{adj}^{2}$	Z	$R_{adj}^{2}$
**0.11**	12.04	0.99	8.86	0.98	11.89	0.72	12.29	0.99	11.85	0.98	10.45	0.99
**0.33**	12.10	0.99	10.84	0.97	9.41	0.91	13.45	0.97	12.40	0.98	9.89	0.99
**0.53**	11.31	0.80	9.23	0.98	11.47	0.96	13.09	0.97	9.69	0.99	9.47	1.00

STy, STe and SAg are the abbreviations for serovars Typhimurium, Tennessee and Agona.

## Data Availability

Not applicable.

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
