# Peer review of "Thermal Death Kinetics of Three Representative Salmonella enterica Strains in Toasted Oats Cereal"

_microorganisms, 2022, doi:10.3390/microorganisms10081570_

Round 1

Reviewer 1 Report

In these study authors focused on the survival of three serovars Salmonella in dry food products. It is only three serovars, two rare (Tenessee and Agona),  and Typhimurium.  Manuscript compared three individual strains were obtained from outbreaks – Tennesee and Typhimurium linked to peanut butter, Agona from toasted oats cereal. This study includes only three serovars, so general statements to all Salmonella, especially Enteritidis are not possible. It should be used additional strains, field or even from ATCC collection. The manuscript presents an interesting study, but not to Microorganism, rather than to journal focused on technical aspects of food production or mathematical models. The results indicated  for some thermal tolerance amongst Salmonella serovars based on mathematical models but nothing is know about this mechanism (succrose adding?)

In my opinion, this article is not written in the scientific standard that characterizes the Microorganisms and should not be published.

Reviewer 2 Report

The authors completed the manuscript and made the necessary corrections.

This manuscript is a resubmission of an earlier submission. The following is a list of the peer review reports and author responses from that submission.

Round 1

Reviewer 1 Report

 The manuscript presents an interesting study with practical applications on the incidence of Salmonella in foods with low values of aw.

 As the food model the authors chose a well-known food, the commercial toasted oat cereal (TOC), which was processed, inoculated with three Salmonella serovars (Agona, Tennessee, Typhimurium) and equilibrated to different values of aw.

The methods used are clearly explained, and the results obtained showed that thermal tolerance is inversely dependent on aw and is influenced by the type of serovar.

Observations:

Line 19 - “108” - 108

Line 118 “The cereal samples were then ground using sterilized mortar and pestle” - the average size of the TOC particles should be specified after grinding, if known

2.3. - it is not clear how much TOC was introduced into the capillary tubes.

In section 3. Results, it would have been more suggestive that some data in the table, for example D values, be plotted against aw and temperature.

In section 4. Discussions it would be interesting to try a possible explanation for the influence of aw on the number of Salmonella cell survivors, based on several processes, such as the irreversible destabilization of ribosomes, accumulation of osmoprotectant molecules, regulated genes, viable but nonculturable state of bacterial cells, effect of coagulation or oxidation of proteins (including enzymes) in the microbial cell.

Reviewer 2 Report

In their study authors focused their research on the survival of Salmonella in dry food products. Authors determine the inactivation kinetics of Salmonella on cereal using three Salmonella serovars and two math models.

As far as recognition of the mechanism by which Salmonella enters and survived the food production chains is needed and important, this particular paper does not answer any question regarding this mechanism.

It is a piece of work that bring a lot of new data regarding Salmonella persistence in dry food, however, do not bring any truly novel information. It is just a change in the experimental model, so what is the unique value of this paper?

It will be useful to present all the experimental data and results in a graph form – to make It easier to follow and more informative.

Reviewer 3 Report

Thermal Death Kinetics of Salmonella enterica in Toasted Oat Cereal

The work has a very interesting premise.

However, it requires significant improvements.

The title of the work is too general because, in fact, three strains of Salmonella enterica subsp. enterica Typhimurium, S. Tennessee and S. Agona  were tested.

I do not understand the strain labels - they should be deposited in the world repository under a specific number.

line 45-46 "Recent epidemiological reports confirmed that in 2015 (...)[15]" - 2016 is not recently, please find the current stats in US https://www.cdc.gov/salmonella/oranienburg-09-21/details.html

line 75-78 The description above the table is enough, there is no need to insert the whole table, the more so that it mentions completely different Salmonella spp serovars than the tested ones.

line 91-92 Salmonella enterica

should be: Salmonella enterica subsp. enterica

Line 109-125

The inoculum solution should contain a certain number of bacteria which should be clearly described.

The microbial contamination of the TOC has not been described, it should be detected and specified.

Line 120

Where did the sucrose come from (company, city, country) was it tested for microbiological purity?

Part 2.3. Preparation of samples for thermal inactivation

It is too general, it should be clear which Salmonella spp in what configuration TOC or / and sucrose and of course in what water activity. The table would be clearer.

For the detection of Salmonella spp in food, the recommended microbiological media are: MSRV, XLD to determine the number of microorganisms there is a specific standard according to which scientists should test e.g. in EU is  EN ISO 6579-1: 2017 Microbiology of the food chain - Horizontal method for the detection, enumeration and serotyping of Salmonella. - even if it is a laboratory experiment

I found a similar description of the experience in "Thermal Inactivation Kinetics of Salmonella Serovars on Dry Cereal" A Thesis Submitted to the Faculty of the Graduate School of the University of Minnesota By Matt Chick In Fulfillment of the Requirements for the Degree of Master of Science Dr. Francisco Diez-Gonzalez, Advisor August 2011 © Matt Chick 2011

Prabably that is why the methodology is not adapted to current standards, because it is a study from 11 years ago, but it is not described in M&M, which reduces the credibility of the entire publication.

Unfortunately, in my opinion, this article is not written in the scientific standard that characterizes the Microorganisms and should not be published in this form.